# Implants Survival Rate in Regenerated Sites with Innovative Graft Biomaterials: 1 Year Follow-Up

**DOI:** 10.3390/ma14185292

**Published:** 2021-09-14

**Authors:** Elio Minetti, Martin Celko, Marcello Contessi, Fabrizio Carini, Ugo Gambardella, Edoardo Giacometti, Jesus Santillana, Tomas Beca Campoy, Johannes H. Schmitz, Mauro Libertucci, Henrykim Ho, Simon Haan, Filiberto Mastrangelo

**Affiliations:** 1Department of Biomedical, Surgical, Dental Science, University of Milan, 20161 Milan, Italy; 2Independent Researcher, 500 02 Hradec Kralove, Czech Republic; doktorcorporal@seznam.cz; 3Independent Researcher, 34121 Trieste, Italy; contegiallots@hotmail.it; 4Department School of Medicine and Surgery, University of Milano Bicocca, 20100 Milan, Italy; fabrizio.carini@unimib.it; 5Independent Researcher, 24068 Seriate, Italy; ugo.gambardella@gmail.com; 6Department of Medical Sciences and Diagnostic Integrated, University of Genoa, 16121 Genova, Italy; edogiac@libero.it; 7Independent Researcher, 48001 Bilbao, Spain; jsantillanaalia@hotmail.com; 8Independent Researcher, 28006 Madrid, Spain; doctorbeca@gmail.com; 9Independent Researcher, 20100 Milan, Italy; drjschmitz@gmail.com; 10Independent Researcher, 00010 Rome, Italy; maurolibertucci@gmail.com; 11Independent Researcher, Singapore 238863, Singapore; Henryhokl1208@gmail.com (H.H.); gt7900@gmail.com (S.H.); 12Clinical and Experimental Medicine Department, School of Dentistry University of Foggia, 71122 Foggia, Italy

**Keywords:** alveolar socket preservation, dental implants, dental implants survival rate, peri-implant bone loss, bone grafts, guided bone regeneration

## Abstract

In thirteen different dental clinics in Singapore, Spain, Czech Republic and Italy, 504 patients were selected, and 483 dental implants were placed in maxillary sites after alveolar socket preservation (ASP) procedures with an autologous demineralized tooth extracted as graft material from an innovative Tooth Transformer device was obtained. All procedures used were reported in n°638 Ethical Committee surgical protocol of University of Chieti and approved. After 4 months, at dental implant placing, bone biopsies were performed to evaluate the histologic outcomes, and 12 months after implant loading, global implant survival rate, failure percentage and peri-implant bone loss were detected. After ASP, only 27 post-operative complications were observed and after 4 months, bone biopsy histomorphometric analysis showed a high percentage of bone volume (BV) 43.58 (±12.09), and vital new bone (NB) 32.38 (±17.15) with an absence of inflammation or necrosis areas. Twelve months after loading, only 10 dental implants failed (2.3%), with a 98.2% overall implant survival rate, nine cases showed mucositis (1.8%) and eight showed peri-implantitis (1.6%). At mesial sites, 0.43 mm (±0.83) of bone loss around the implants was detected and 0.23 mm (±0.38) at the distal sites with an average value of 0.37 mm (±0.68) (*p* > 0.568). Several studies with a longer follow-up will be necessary to confirm the preliminary data observed. However, clinical results seem to suggest that the post-extraction socket preservation procedure using innovative demineralized autologous tooth-derived biomaterial may be a predictable procedure to produce new vital bone able to support dental implant rehabilitation of maxilla edentulous sites.

## 1. Introduction

After tooth extraction, the maxillary alveolar bone undergoes several physiological or paraphysiological changes with a constant volumetric bone loss. Several surgical and prosthetic complex procedures have been proposed to contrast the bone remodeling and loss to avoid the failure of dental implant rehabilitation, especially in aesthetic areas.

Several studies on guided bone regeneration (GBR) with different biomaterials have been performed to neutralize or reduce the bone loss and promote bone healing and growth in post-extraction socket sites.

In 2016, the World Health Organization (W.H.O.) determined that edentulism wasstill a major public worldwide health problem and considered single or multiple edentulism as a functional and psychological disability for millions of persons [1]. Currently, dental implant rehabilitation is considered the standard treatment to solve tooth loss with a high success rate [2,3,4,5]. However, after tooth extraction, the alveolar bone undergoes several physiological changes which produces volumetric maxillary bone loss [6]. However, several condition play a crucial role for the dental implant prognosis and success; first of all were the qualitative and quantitative bone conditions where the dental implants were placed [7,8,9].

Alveolar bone volumetric changes after tooth extraction are considered a crucial challenge in implant dentistry [10,11,12,13,14,15].

Several autologous bone grafts from different donor sites and allogenic, xenogenic and synthetic heterologous graft biomaterials were surgically used to counteract the alveolar bone resorption, promoting ideal condition for implant osseointegration. Autologous bone graft is yet considered the gold standard material for alveolar socket preservation (ASP) or guided bone regeneration (GBR) procedures for osteoconduction and osteoinduction capabilities without immune rejection [16,17].

In the last ten years, several in vitro and in vivo studies have been developed to understand the right surgical procedures and to know the best autologous or eterologous biomaterial to reduce and to regenerate bone loss [18,19,20,21,22,23].

In any case, autologous bone graft showed a high donor site morbidity, limited availability and high resorption rates. For this reasons, several heterologous graft biomaterials were clinical used with osteoconduction capability and high cost and potential disease transmission disadvantages. Several studies were promoted to find the ideal biomaterial for bone regeneration [24].

In 2012, Jung RE et al. [25], in a systematic review and meta-analysis, showed a 97.2% dental implant survival rate after 5 years and 95.2% after 10 years.

In 2017, Compton SM et al., in an elderly population, observed 92.9% overall implants survival rate and 7.1% of failure and marginal bone loss in 23.3% of the implants in a long-term follow-up review [26]. 

In 2019, Howe MS et al. showed a 96.4% survival implant level after 10 years [27]. In 2019, Nassani MZ, in a follow-up period ranging from one to five years, showed a 86.3% mean survival rate of replacement dental implants and that initial implant failures are mostly related to modifiable risk factors [28].

In 2020, Alghamdi HS et al. reported the survival rate of dental implants above 90%, and this value was related to compromised bone conditions promoting implant failure and endangering the current high success rates [29].

In 2020, Oh SL et al. after 1–5 years follow-up confirmed the cumulative mean survival rate of implants was 86.3% after retreatment. The survival rates in 217 retreated implants revealed a significantly higher value for rough-surfaced implants than for smooth-surfaced implants (90% versus 68.7%) [30].

In 2000, Manz et al. [31] found that 1405 implants placed in natural bone showed 0.4 mm (±0.7) bone loss mean value after 12 months, and in 2006, Cardaropoli et al. detected 1.8 mm (±0.7) bone loss around the dental implant after only 1 year. In 2016, Apostolopoulos et al. [32], found an average of 3.45 mm (±0.80) bone loss around 51 implants in regenerate sites with xenograft, and 2.48 mm (±0.63) bone loss around 42 implants in healing sites without graft material. In 2017, Maiorana et al. [33], using the same materials, showed 1.21 mm (±0.46) mean width and height bone resorption.

In 2020, Crespi et al. [34], in a study with 42 implants, 19 after GBR using a magnesium-enriched hydroxyapatite, 23 after GBR using porcine bone, and 21 implants in nongrafted bone, reported a success rate of 88.1% dental implant survival rate in sites after alveolar socket preservation procedures with graft materials and 85.7% implant success rate in maxillary sites without grafting after 10-year follow-up.

Recent human study has shown histological and hystomorphometrical high levels of vital bone formation in ASP or maxillary sinus floor augmentation (MSFA) procedures, using autologous demineralized dentin matrix grafts from fresh extracted teeth obtained with a Tooth Transformer innovative device [35,36,37,38].

The aim of the present multi-center retrospective study was to evaluate the bone loss around 483 dental sites, the survival rate and predictability of dental implants placed in human maxillary sites after Alveolar Socket Preservation (ASP) procedures through an innovative autologous graft material derived from fresh extracted teeth, 1 year after loading.

## 2. Materials and Methods

### 2.1. Study Design

The primary outcome of the prospective study was to evaluate the surgical intra or post-operative complications of the implant insertion in alveolar sockets preservation sites obtained with tooth-derived innovative autologous tooth grafts.

The secondary outcome was to analyze the success rate (cumulative—pre-loading or after-loading, implant failure, prosthetic failure) of the dental implants after 1 year of follow-up in alveolar socket preservation sites obtained with tooth-derived innovative autologous tooth grafts.

The third outcome was to evaluate the survival rate of the dental implants after 1 year of follow-up (bone loss, mucositis, perimplantitis) in alveolar socket preservation sites obtained with a tooth-derived innovative autologous tooth graft. Incidence of peri-implantitis of the implants under maintenance after one year was calculated based on the Consensus report of the 4th workgroup of the 2017 World Workshop [39].

Bone loss beyond the reference point was calculated by comparing peri-apical radiographs taken during recall visits after regeneration, after the implant insertion and after 1 year with baseline (implant installation). The mean of both bone level readings (mesial and distal) was calculated to obtain a peri-implant single value.

Implant failure was evaluated when the dental implant was removed due to the following reasons: implant mobility, loss of osseointegration, ongoing bone loss, infection, persistent pain or patient discomfort.

Implant success was evaluated when total bone loss beyond the reference point, from the placement of the implant to 1 year of follow-up, was less than 2 mm.

On 21st March 2019, the University of Chieti Ethics Committee authorized the clinical study protocol on a human model registered at the number: 638—21/3/19. Between April 2019 and December 2020, in 13 different dental clinics in Italy, Spain, Czech Republic and Singapore, five-hundred and four patients (235 male and 269 female) in different health conditions, with an average age 54 yrs (range 22–85 yrs), were recruited (Table 1). Twenty-one recruited patients have not completed the protocol. Six of them did not show up at the appointments. Five of them died. Seven of them have moved and are no longer under control of protocol centers and three of them were excluded for illness (cancer).

Preliminary clinical and radiographic evaluation and written informed consent have been performed. For different reasons, all patients needed tooth extraction. (Table 1) Two weeks before surgery, all patients received a professional oral hygiene session. Patients with allergies, smoke, healing disorders, pregnancy, diabetes, cancer, HIV, bone diseases, metabolic diseases, systemic corticosteroids use, intramuscular or intravenous bisphosphonates use, immunosuppressive agents use, radiation therapy and chemotherapy were excluded. Tooth extractions were performed, and inflammatory tissue was completely removed. After tooth extraction, in all maxillary sites the alveolar socket preservation procedures were performed with fresh demineralized autologous graft from TT Tooth Transformer device (Milan, Italy) and covered with a resorbable collagen membrane. This device, the procedure and the advantages of this procedure were described in a recent book [40]. A suture closed the flap and secured the membrane in place. After four months, 100 bone biopsies were performed to histological evaluation and 483 dental implants were inserted. (Figure 1) Data statistical analysis was carried out to obtain average values and to compare the differences between bone loss around the implants. Outcome measures of the exploratory study were analyzed with a t-test for paired pre–post differences with time as the factor, using the Statistical Package for Social Sciences (SPSS for Windows, Version 11.5, Chicago, IL, USA) software to detect significant differences between pre-test and post-test bone loss around the implants.

### 2.2. Surgical Procedures and Follow-Up

After gentle tooth extraction, the baseline socket buccal-lingual and vertical morphology dimensions were recorded. The whole extracted tooth was cleaned using a diamond drill (ref.6855 Dentsply Maillefer, Ballaigues, Switzerland) with abundant water irrigation and any filling materials (gutta-percha, composite, etc.) were carefully removed from the tooth. After, the tooth was cut in small pieces (10 × 10 mm) and inserted in the milling device (Tooth Transformer, Milan, Italy). According to the manufacturer, a single-use box containing disposable liquid solutions was inserted in the device to guarantee the decontamination of the grafts. After 25 min, particle graft biomaterials were obtained, placed in the alveolar post-extractive socket sites and finally covered with a resorbable collagen membrane. The graft material was confined to the existing alveolar ridge dimensions, making no attempt to go outside the confines of the ridge. Amoxicillin (Pfizer, New York, NY USA) was provided (1 g × 2 × 7 days). After 2 weeks post-surgery, the sutures were removed and, after the 4-month healing period, 100 bone biopsies using 4 × 18 mm graduated trephine cylindrical drills (Meisinger, Neuss, Germany) were performed and 483 dental implants (CEA, Heckermann, 3i, Dentium, ICX, BTI, Prodent, Straumann) were placed in alveolar regenerated sites.

### 2.3. Histological Technique

All samples were washed, dehydrated with increasing concentration alcohol solutions (Sigma-Aldrich, St. Louis, MO, USA), and then infiltrated into methacrylic resin (Sigma-Aldrich, St. Louis, MO, USA) for the histological analysis. After, the sample was processed to obtain non-decalcified sections using a disk abrasion system (LS2—Remet, Remet, Bologna, Italy) and diamond disk cutting system (Micromet—Remet, Bologna, Italy) to obtain sample slides about 200 microns thick. Then, all samples were treated with low abrasive paper on the lapping machine (Bueheler, Lake Blu, IL, USA) with thickness control that allows for progressively reducing the sample thickness up to about 40–50 microns. Finally, the specimens were polished, colored with basic fuchsin and blue toluidine and observed with light and polarized light microscopy (Olympus, Shinjuku, Tokyo, Japan). The histological images obtained from the transmitted light microscope (Olympus, Shinjuku, Tokyo, Japan) were digitized through a digital camera and analyzed by means of an image analysis software IAS 2000 (QEA, Billerica, MA, USA). For each sample, percentage of residual bone volume with exclusion of medullary tissues (BV%), = percentage of the remaining graft, excluding bone and marrow (Graft%), and vital bone percentage with exclusion of the medulla and residual graft (VB%) were detected.

### 2.4. Complications

Complications were classified in early and late post-surgical complications. The early included wound dehiscence, hematoma and edema, while late included peri-implant mucositis, peri-implantitis and prosthetic complications.

### 2.5. Patient’s Satisfaction

All patients filled in a customized flow chart with the aim to understand the patient satisfaction about this procedure. All of the patients were enthusiastic to use their own tooth for regeneration and the flow chart questions were aimed at understanding not only satisfaction, but also if this type of regenerative could have a lower morbidity. One week and 1 month after surgical procedures, the patient morbidity, swelling and pain flowchart was recorded. Patients’ satisfaction and whether they were willing to repeat the same surgical procedure were recorded after surgery and at the time of implant loading. 

### 2.6. Failure

Failure was classified in early post-surgical failure, early or late post-implant insertion failure and post-loading failure.

## 3. Results

### 3.1. Peri-Implant Bone Loss Evaluation

After the surgical procedures (baseline) and 1 year after the prosthetic loading of the dental implants, peri-apical X-Ray and graduate periodontal probing were obtained to evaluate the implant osseointegration, the bone condition and resorption, the mesial-distal bone level, mucositis and peri-implatitis. (Table 2) The implant–abutment junction was arbitrarily chosen as a reference point. The distance between this reference point and the marginal bone level was measured in millimeters at the mesial and distal site of each implant. Two independent examiners, not related to the surgeons, analyzed all radiographs twice. Nineteen randomly selected radiographs were measured two-fold to analyze intra-examiner reproducibility.

### 3.2. Asp Implant Failure and Complications

After ASP procedures, five early post-operative complications were recorded: three wound dehiscence and two hematomas. A total of ten implant osseointegration failures were observed and seven implants were lost in the same dental clinic before loading. Seven failed at pre-loading time during the healing period, two at 2 months after loading and one at 4 months after loading (Table 2). Three were in the mandible and seven in the maxilla. Seven defects were with two walls and three defects were with three walls. After loading, 1.8% of mucositis (9/483) and 1.6% (8/483) peri-implantitis were detected. All complications were treated with local medication, surgical curettage and new suture.

The failed dental implants were removed and, after 3 months bone healing time, another implant fixture was inserted in the same alveolar site. No prosthetic failure was observed during the follow-up period. In January 2019, one hundred and five implants were inserted and two failures, two instances of mucositis and two of peri-implantitis were recorded. In February 2019, one hundred and fifty implants were inserted and four failures, three cases of mucositis and three of peri-implantitis were recorded. In January 2020, one hundred and sixty-two implants were inserted and four failures, three cases of mucositis and two of peri-implantitis were recorded. In February 2020, sixty-six implants were placed, with zero failure, one case of mucositis and one of peri-implantitis recorded. Twelve months after the insertion of 483 implants and after the loading, the cumulative implant failure rate was 1.75% and the implant survival rate was 98.2%. (Figure 2).

The number of clinical cases treated in each center is presented in Figure 3.

### 3.3. Histological Analysis

After a 4-month ASP healing period, during dental implant placement surgical procedures, 100 bone biopsies were obtained and the specimens’ histomorphometric analysis showed 43.58% (±12.09) average value of BV (bone volume). In all samples, the average rate of RG (residual graft) showed 10.47% (±10.68) and the 32.38 (±17.15) of NB (new bone) was detected. In all samples, no inflammation or necrosis area, and no filling endodontic materials were detected. In all specimens, dentin and an enamel residual matrix were observed. (Figure 4)

### 3.4. Peri-Implant Bone Loss

One year after dental implant loading, mesial-distal bone levels around dental implants were evaluated with peri-apical X-ray (Figure 5) and graduate periodontal probing. After 1 year, the average mesial bone loss was 0.43 mm (±0.83) and distal bone loss was 0.23 mm (±0.38). A total bone loss average value was 0.37 mm (±0.68).

### 3.5. Patient’s Satisfaction Rate

A total 90% of the patients showed satisfaction for the treatment. No morbidity in 79.5%, no swelling in 78%, no pain in 15%, moderate pain in 75% and high pain level in 10% were recorded. (Table 3) An important indication derived from this analysis is that the use of the autologous tooth is considered in a highly positive manner by patients. The patients themselves have presented a reduced morbidity, and also in case of further regeneration would want to use one of their teeth again.

## 4. Discussion

In the present study, the results showed high tridimentional bone volume and high percentage of vital bone formation in all ASP sites with a low grade of bone loss around the dental implant 1 year after loading, compared with the literature.

After one year of follow up, in the present study, only 10 dental implant failures were observed with a global 1.75% rate. After 2 months, two implants failed, and only one failed 4 months after loading. Mucositis after loading was observed in 1.8% of patients (9/483) and peri-implantitis was detected in 1.6% (8/483). Twelve months after ASP procedures, implants insertion and loading, a 98.2% (473/483) cumulative implant survival rate was detected.

After 4 months, the histological analysis of one hundred alveolar sockets preservation site biopsies showed no necrosis or inflammation area or filling with endodontic material in all samples analyzed. The biopsies’ histomorphometric analysis showed a 43.58% (±12.09) average value of bone volume, 10.47% (±10.68) of dentin/enamel residual graft and 32.38 (±17.15) of the new bone.

After a one year follow-up, the data confirmed a lower failure rate and an higher dental implant survival rate compared with the literature.

The use of teeth as grafting material is a relatively new field of research.

The tooth, consisting of Hydroxyapatite, type 1 collagen (the same as bone) and a small amount of non-collagen proteins, all autologous origin, suggests a perfect material for regeneration.

Common grafting materials are mainly composed of hydroxyapatite of various origins, and lately some have a collagenation. However, what is their origin?

In an increasingly bio-oriented world, a bio choice in how to use the autologous tooth changes the approach of patients towards regenerative therapies. This choice must be combined with the undoubted high quality of tooth regeneration.

Acceptance of these therapies, as can be seen from the satisfaction score, is almost 100% and also includes patients who follow vegan food choices.

## 5. Conclusions

Further studies must be promoted with a large number of implants, and a long observation time could be needed to evaluate and understand the real impact of the demineralized tooth graft materials in bone regeneration of oral and maxillofacial procedures.

However, the high success rate of dental implants one year after loading in regenerated sites with extracted teeth confirmed the high biocompatibility of this innovative autologous graft material. The high bone volume and vital bone value in all samples analyzed confirmed an important role of the extracted tooth, not as a waste, but as an innovative graft material in ASP procedures. The human demineralized dental graft material high stability, the low bone loss level around the dental implant and the high survival rate of dental implants 1 year after loading confirmed the potential role of the tooth used in intra-oral bone maintenance, preservation and augmentation treatment.

## Figures and Tables

**Figure 1 materials-14-05292-f001:**
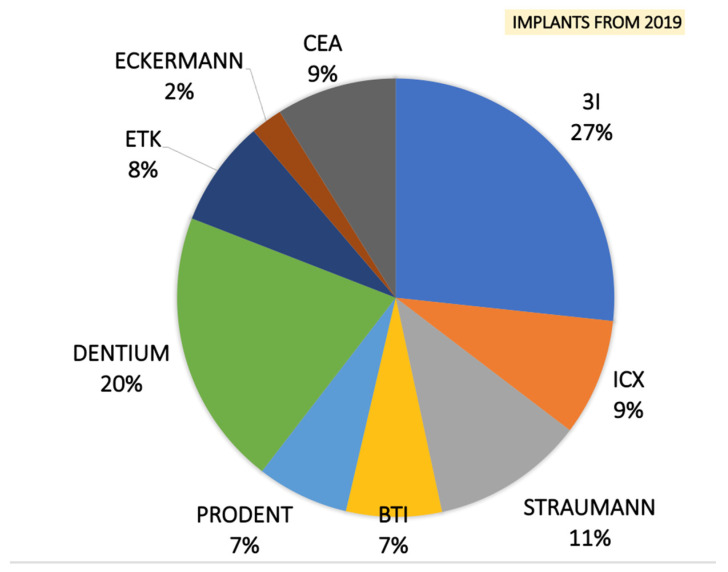
Four months after alveolar socket preservation procedures, a total of 483 dental implants were placed from 9 different dental implants systems.

**Figure 2 materials-14-05292-f002:**
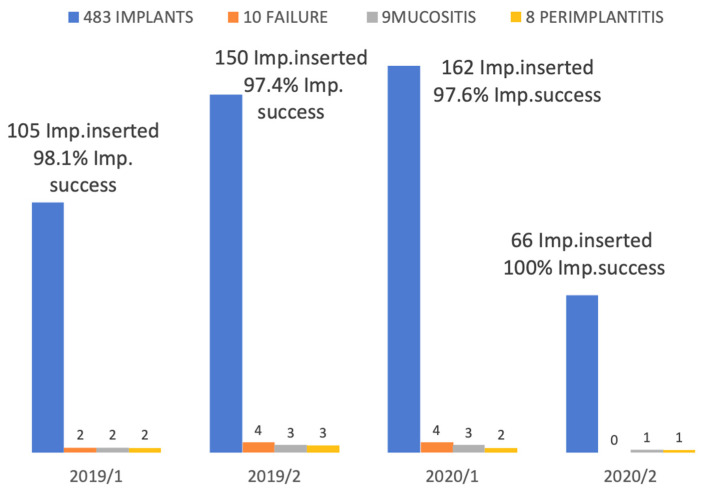
Numbers of implants inserted in ASP sites. success, failure, mucositis and periimplantitis rate detected.

**Figure 3 materials-14-05292-f003:**
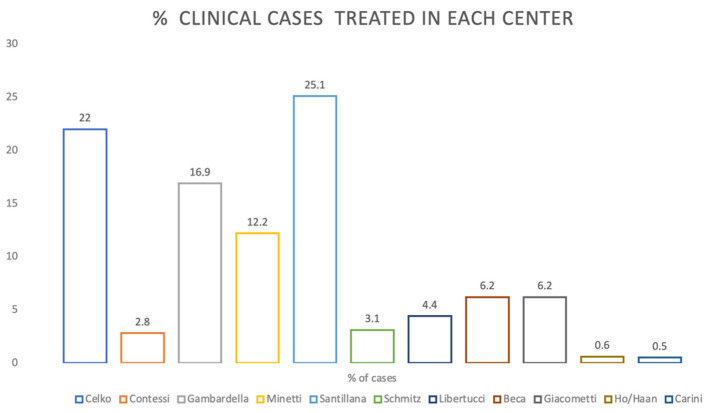
Percentage of cases in each center.

**Figure 4 materials-14-05292-f004:**
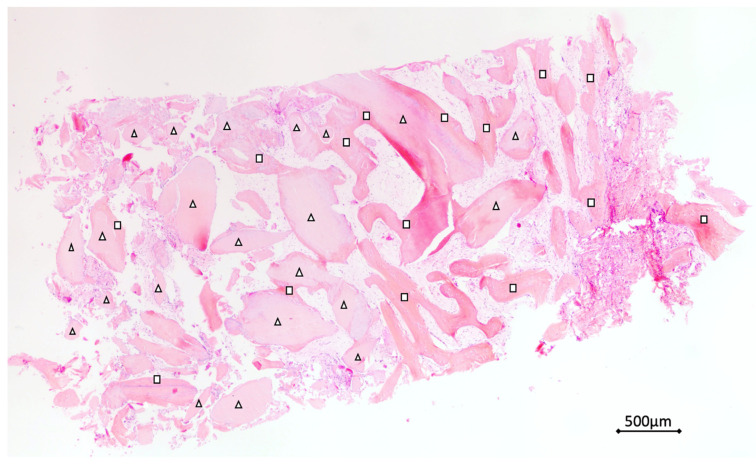
Biopsies istological and histomorphometrical analysis. (Hematoxylin and eosin colored). Total bone volume (BV), residual graft material (RG) and new bone (NB) were evaluated. With the triangle, the dentin/enamel granules (RG), and with the square the new bone (NB) were indicated.

**Figure 5 materials-14-05292-f005:**
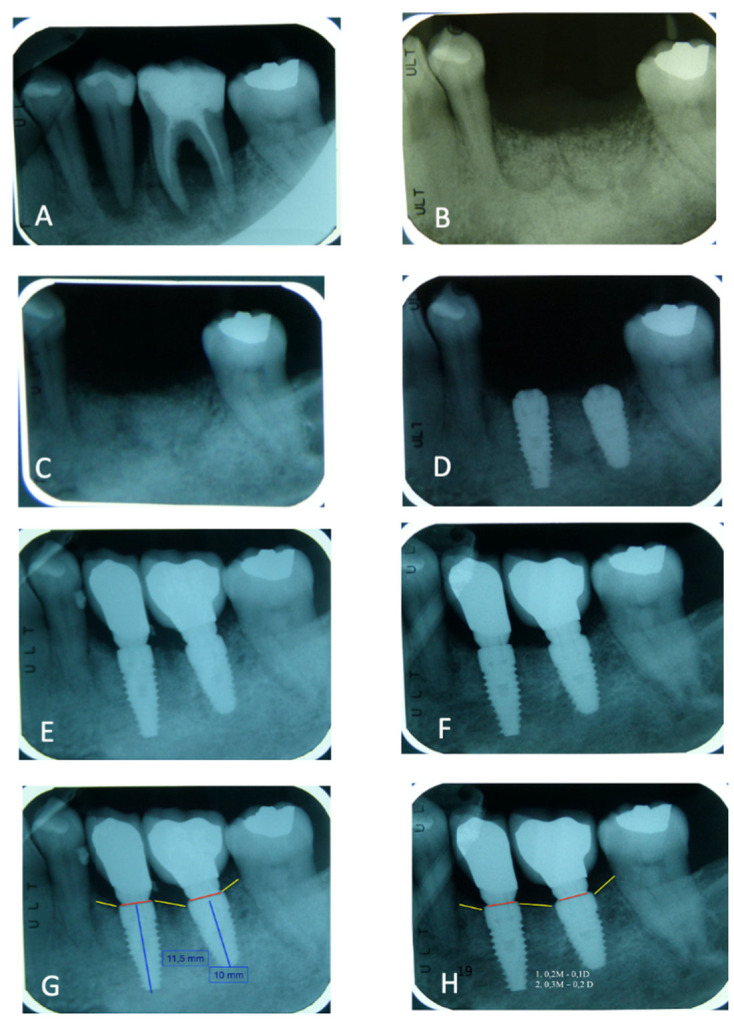
(**A**) Elements 3.5 and 3.6 before the extractions—(**B**). ASP immediately after the surgery—(**C**). Radiographic control after 3 months—(**D**). Implant insertion after 4 months—(**E**). Image after the prosthetic load—(**F**). Image after 1 year of load—(**G**). Bone level around dental implants after the load—(**H**). Average mesial-distal and total bone loss value were detected after one year of prosthetic load.

**Table 1 materials-14-05292-t001:** Between April 2019 and December 2020, in 13 different dental clinics in Italy, Spain, Czech Republic and Singapore, 504 patients were treated with 483 dental implant rehabilitations.

Dental Clinic Surgeons	13
Sample Size	504 (269 Female–235 Male)
Average Age	54.09 (Range 22–85)
Total Extracted Teeth Incisive Canine Premolars Molars	524 Incisive 104 Canine 42 Premolars 184 Molars 194
Extraction Reasons	Crown Trauma Eternal Root Resorption Periodontitis Root Fracture Infected Root
Total Socket Sites Treated	483 (Maxilla 278–Mandible 205)
Dental Implants	483
Average Implant Length	10.83mm (±1.17)

**Table 2 materials-14-05292-t002:** Implant failure: partition with site, failure rate, implant dimensions and type of implants lost.

**IMPLANT FAILURE**		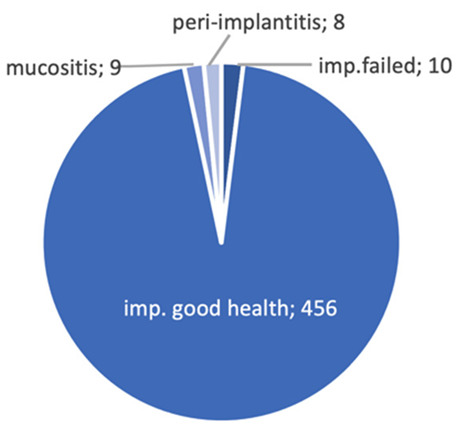
Site where the implants were failed	23–12-14–25–25–21-26–46–34–36	
Implant failure rate	1.75%	
Number of failed implants for each dimensions	N°2 (4 × 8.5 mm) N°1 (4 × 10 mm) N°1 (5 × 13 mm) N°2 (4 × 11 mm) N°1 (3.75 × 10 mm) N°2 (4.1 × 12.5 mm) N°1 (6 × 11 mm)	
Number of implants lost for each producer	N°7 3I N°3 ICX	

**Table 3 materials-14-05292-t003:** Patient’s satisfaction schedule value.

PATIENT SATISFACTION SCORE			
	YES	YES/NO	NO
Satisfaction to use tooth as graft	90%	10%	
Morbidity	5.5%	15%	79.5%
Swelling	10%	12%	78%
Would you do it again?	88%	12%	

## Data Availability

The data underlying this article will be shared on reasonable request from the corresponding author.

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
