# Peer review of "Implants Survival Rate in Regenerated Sites with Innovative Graft Biomaterials: 1 Year Follow-Up"

_materials, 2021, doi:10.3390/ma14185292_

Round 1

Reviewer 1 Report

You can find attached the PDF file.

Author Response

dear reviewer

1- we move the sentence at the end of the introduction

2-we add

3-all the table and figures have a description, but suggest you a different description? or an extension of the text?

4- ok we add

5- 504 are the patients recruited for this study but some of them  aren't end the protocol. the data are made using only the 483 dental implants rehabilitations because they are end the protocol. but we add this data on the paper

6-7-8- what do you means exactly. do you need additional informations about implants or graft materials?or do you suggest to check the spelling?

9- I agree but if all the implants types are in good conditions..the graft produce a good support for all implants

10- ok we check the table

11-ok

12-we check

13- ok we check

14- ok we check

15- we indicate this data because  probably the clinic had a bad surgery in comparison with the others clinics 

Reviewer 2 Report

In this study the authors have evaluated the peri-implant bone loss, the survival rate and predictability of dental implants placed in human maxillary sites 1 year after Alveolar Socket Preservation (ASP) procedure using an autologous graft material derived from fresh extracted teeth. While the idea of using fresh extracted teeth as the graft material seems interesting, the conduct of the research makes it difficult to justify the results of the study. The main flaw of the study is the lack of any control groups (negative or positive controls) which makes it impossible to assess the pure effect of employed material irrespective of other confounding factors such as surgical factors, patient specific parameters etc.

The other issue is the lack of any calibration among the clinicians regarding the surgical procedure as well as the clinical assessment of the cases after the surgery. This issue is of more significance considering that the operations have been performed in multiple centers in different countries.

Author Response

dear reviewer 

this article  is a retrospective study . We made this tipe of study to evaluate the implants survival rate after 12 months inserted in regenerated bone using autologous teeth.

the calibration among the clinicians regarding the surgical procedure was made with a Protocol. I send our protocol to the editor.

kind regards

Reviewer 3 Report

The manuscript entitled "IMPLANTS SURVIVAL RATE IN REGENERATED SITES
WITH INNOVATIVE GRAFT BIOMATERIALS: 1 YEAR FOLLOW-UP" by Minetti et al. describes the clinical data collected from 13 different dental clinics including 504 patients and 483 dental implants to study the regenerative abilities of a novel graft biomaterial placed in maxillary sites after alveolar socket preservation (ASP) procedures with autologous demineralized tooth extracted as graft material. The results from this analysis are promising and suggest that the postextraction socket preservation procedure using innovative demineralized autologous tooth-derived biomaterial may be beneficial to patients in producing new vital bone that can support dental implant rehabilitation. 

The study is well-designed and executed. However, there are minor concerns as listed below:

Figure 1 is of low magnification. An additional enlarged view of insets from this figure will be helpful. Scale bars need to be incorporated into the original figure.

Table 2, Table 3, and Table 4: The font size in this diagram are too small to visualize. Please increase the font size in this figure.

Except for figure 1, no other figure or table included the description of figure legend. The authors need to include a short description of the figure or table below the figure legend.

The section 2.11: Results from the Patient satisfaction rate analyses is not described well. The authors need to elaborate this section.

The discussion is poorly written. The authors have cited some previous references and narrated the key results of some previous publications. However, they failed to explain how the results from their study relate to those that have tried previously with different other biomaterials and other approaches. The authors should highlight the advantages of their approach with reference to those that have previously tried and failed or have reported limited success rates. The discussion should go in this direction to enable the readers to understand the progress in the field and how their work has implications for the future.

Author Response

dear reviewer

thank you for your comments

fig 1. yes I contact the histologist to have a better image

fig 2 ok we increase the font

except the fig 1 in the figure we have a short description...do you suggest to increase?

section 2.11 we modify this section

about the discussion we modify 

Reviewer 4 Report

Dear authors,

According to my peer review, the following manuscript entitled - Implants survival rate in regenerated sites with innovative graft biomaterials: 1 year follow-up addresses a relevant topic and may fall within the scope of Materials.

Unfortunately, and considering the failing aspects mentioned above I think your manuscript text should not be accepted for publication:

1) This manuscript focus its relevance by testing a "innovative" (according to authors) biomaterial in alveolar ridge preservation. However, in my opinion, the methodology employed completely fails to provide crucial aspects regarding the potential of this biomaterial in alveolar ridge preservation. Without a control group - spontaneous healing - the information provided by this manuscript is scarce and weaken the quality of this study in order to validate the use of this biomaterial. As a reviewer, the following question should be formulated: how different would be the outcomes measured if the alveolar sockets were left in uneventful healing.

2) Considering the following information: "After tooth extraction, in all maxillary sites the alveolar socket preservation procedures were performed with fresh demineralized autologous graft from TT Tooth Transformer device". Missing information related with: how innovative is this technique since it is poorly described?/ how does this device works?/ how mineralised tissues are processed?

3) For bone sample collection authors propose a graduated trephine 4x18 mm graduated trephine cylindrical drills. The depth of bone collection is not mentioned in the manuscript. Was it equal for all centers? How to use this trephine diameter without impairing higher implant primary stability?

4) How apico-coronal direction of the bone sample was indicated in order to interpret histologic results?

5) How many specimens were obtained from each bone sample? In just one part of the bone cylinder (apical?coronal?)

6) Histological and histomorphometric analysis lacks quality definition.

7) "One week and 1 month after surgical procedures, the patient morbidity, swelling and pain flowchart was recorded. Patients' satisfaction and whether they were willing to re-peat the same surgical procedure were recorded after surgery and at the time of implant loading.

Which type of flowchart was used? Based on VAS scale?

8) Being a multi centric study, number of clinical cases treated in each center is also missing.

Author Response

dear reviewer

1 -we respect the reviewer opinion but the manuscript not show the result of the gbr that were published in different papers focused to asp procedures where the opinion of the reviewer were analysed and the quality of the regeneration was validate . references 7-8-9-10

2-sure we add all these informations 

3- the depth of the sample collection was 10 mm. the minimum implant diameter was 4.2  

4-our interest is not to understand if have a major quantity of bone in depth or in surface. the histomorphometry is a average value.

5- each bone sample was analysed in one section. the sample was selected at the middle point of the asp.

6- the image of the histological sample is a .png 2246x1316 follow the editor indications. is possible to use tiff or jpg to have better resolution. the histimormetric data are numbers. do you need a major quantity of this data?

7-we use a customised flow chart. do you need a copy?

8-ok we add all these data in the paper

Round 2

Reviewer 2 Report

I would like to thank the authors for their revision of the manuscript. However the key issue of the study which is the lack of any negative or positive control groups still exists and the effect of operator bias including the surgical method cant be eliminated from the study results. Accordingly the results of the study can be significantly unreliable and therefore I wouldnt suggest accepting the paper for publication in its current form.

Author Response

The authors would to thank the reviewer for his contribution.
However, the authors would like to reiterate that there are different types of research and there are not only comparative studies in scientific literature.
On the contrary, the present study is one of great scientific depth because it is a clinical research on the patient carried out after acceptance by the ethics committee of the University of Chieti-Italy and under his supervision.
The aim of the study was to evaluate if the tooth was an adequate  autologous biomaterial to bone regeneration, and if the regenerated bone was adequate to support dental implant rehabilitation.
Probably the reviewer confused the purpose of the research by thinking that the study was a research to compare the dental implant survival rate in regenerated bone versus native bone, or to compare implant survival rate in regenerated bone sites with different biomaterials, or to evaluate the delayed or post-extraction dental implants survival rate in regenerated with single biomaterial, or again, to compare the insertion of 2 implants in the same patient to evaluate the survival of the implants in a regenerated vs non-regenerated bone site.
Each reviewer must criticize if there are problems in the management of the study, if there are discrepancies between the purpose of the work and the results obtained, or if there are omitted data, or if the authors did not comply with the proposed protocol, and other design biases or if there are problems during the realization of the study, on the contrary, he should not criticize the architecture of the study itself which instead is extremely innovative, as for the first time it was possible to evaluate the dental implants survival rate in regenerated sites with an autologous biomaterial (patient's tooth) being able to also analyze if and how many bone has been produced in regenerate sites using the histologies specimens analyzed after the bone biopsies obtained during the second stage of the surgical procedures.
The authors want to emphasize that there are no studies on this topic currently in the literature.
The authors also stated in the study that standard procedures were used in the recruitment of patients, in the phases of surgery and that the device used is currently marketed and all days used in all over the world.
For the scientific community, multicentre studies are considered the higher quality research as following a rigid protocol by the operators approved by an Ethics Committee, they show results that happens worldwide in oral surgery and implantology clinics every day in all patients.
Therefore, the reviewer cannot define the results obtained in this study as incorrect and inadequate results and, above all, the authors define unacceptable the comments of the reviewer because they have followed the research protocol with great seriousness approved to the Ethic Committee.
Therefore, the authors ask the editorial board to accept the manuscript also in light of the timely responses and improvements that have been made to the paper and the comments of the other reviewers.